# Zirconia Toughened Alumina-Based Separator Membrane for Advanced Alkaline Water Electrolyzer

**DOI:** 10.3390/polym14061173

**Published:** 2022-03-15

**Authors:** Muhammad Farjad Ali, Hae In Lee, Christian Immanuel Bernäcker, Thomas Weißgärber, Sechan Lee, Sang-Kyung Kim, Won-Chul Cho

**Affiliations:** 1Department of Advanced Energy and System Engineering, Korea University of Science and Technology (UST), 217 Gajeong-ro, Yuseong-gu, Daejeon 34113, Korea; farjadali5@gmail.com; 2Hydrogen Research Department, Korea Institute of Energy Research (KIER), 152 Gajeong-ro, Yuseong-gu, Daejeon 34129, Korea; mymilkoo@kier.re.kr (H.I.L.); lsc1126@kier.re.kr (S.L.); ksk@kier.re.kr (S.-K.K.); 3Fraunhofer Institute for Manufacturing Technology and Advanced Materials IFAM, Branch Lab Dresden, Winterbergstraße 28, 01277 Dresden, Germany; christian.bernaecker@ifam-dd.fraunhofer.de (C.I.B.); thomas.weissgaerber@ifam-dd.fraunhofer.de (T.W.); 4Department of Future Energy Convergence, Seoul National University of Science & Technology, 232 Gongreung-ro, Nowon-gu, Seoul 01811, Korea

**Keywords:** water electrolysis, alkaline water electrolyzer, zirconia-toughened alumina, porous separator membrane, Zirfon^®^ PERL separator

## Abstract

Hydrogen is nowadays considered a favorable and attractive energy carrier fuel to replace other fuels that cause global warming problems. Water electrolysis has attracted the attention of researchers to produce green hydrogen mainly for the accumulation of renewable energy. Hydrogen can be safely used as a bridge to successfully connect the energy demand and supply divisions. An alkaline water electrolysis system owing to its low cost can efficiently use renewable energy sources on large scale. Normally organic/inorganic composite porous separator membranes have been employed as a membrane for alkaline water electrolyzers. However, the separator membranes exhibit high ionic resistance and low gas resistance values, resulting in lower efficiency and raised safety issues as well. Here, in this study, we report that zirconia toughened alumina (ZTA)–based separator membrane exhibits less ohmic resistance 0.15 Ω·cm^2^ and low hydrogen gas permeability 10.7 × 10^−12^ mol cm^−1^ s^−1^ bar^−1^ in 30 wt.% KOH solution, which outperforms the commercial, state-of-the-art Zirfon^®^ PERL separator. The cell containing ZTA and advanced catalysts exhibit an excellent performance of 2.1 V at 2000 mA/cm^2^ at 30 wt.% KOH and 80 °C, which is comparable with PEM electrolysis. These improved results show that AWEs equipped with ZTA separators could be superior in performance to PEM electrolysis.

## 1. Introduction

Power-to-gas (PtG) technology can address the continuing and extensive energy storage issues along with the minimization of CO_2_ emissions. The PtG process changes electric power into chemical energy for steady storage of energy at great length. This system can be very useful in the betterment of energy-related systems in the future [1,2,3,4]. For the implementation of renewable energy on large scale, a cost-effective and energy-efficient water electrolysis system is required [5]. Similar to water electrolysis, energy storage devices also reduce the emission of greenhouse gases and mitigate the issues of global warming. Energy storage devices such as batteries and capacitors are also a hot topic in the energy sector these days owing to their ease of energy storage and usage according to the preference of users. However, for energy storage devices, the electrode design is very important, and it is a very demanding job to design it [6,7,8,9,10,11,12].

Currently, hydrogen generation through water electrolysis is a suitable choice for the multi gigawatt storage of electrical energy from different irregular sources of energy such as solar and wind. Water electrolysis can easily change surplus electricity into useful hydrogen with greater flexibility [13]. The future of renewable hydrogen production mainly depends on polymer exchange membrane (PEM) and alkaline water electrolysis (AWE) technologies [14,15]. The PEM exhibits high current density (2 A cm^−2^ at ~2 V), low ohmic losses, large partial load range, flexibility, swift response, and low overpotential and produces highly pure hydrogen. This high current density allows the PEM electrolyzer to work under a wide range of power outputs [16,17]. However, it has some gas crossover issues that need to be addressed for better performance of electrolyzer [18]. Additionally, the use of expensive, noble-metal-type catalysts and titanium-based current collectors makes this system less cost effective [19,20]. 

On the contrary, alkaline water electrolyzers (AWEs) exhibit higher durability, simplicity, robust performance, low capital cost, and adequate compatibility with non-noble metal catalysts [21,22,23,24,25]. AWEs are in commercial use for quite a long time and are considered a mature technology [26]. However, they exhibit low operating pressure and limited partial load range, slow response time toward dynamic operations, and most importantly, low current densities (below 800 mA/cm^2^ at 1.8 V), owing to the higher values of ionic resistance of the separator membrane and the lower kinetics caused by the utilization of non-noble type catalysts [27,28,29]. These lower densities increased the number of stacks required to obtain enough hydrogen [30,31,32]. Another reason for the low operating pressure and low partial load range is the non-negligible permeability of dissolved gas in the liquid electrolyte i.e., KOH through the separator membrane [33,34]. When the membrane suffers from high gas permeability, it causes high gas crossover issues and makes the operation difficult at the high-pressure level. However, this can be overcome by adjusting the differential pressure between the cathode and anode carefully at the operational level. High gas permeability also leads to oxygen diffusion onto the cathode chamber and reduces the efficiency of the electrolyzer. This can be minimized by developing a membrane with low gas permeability, which can limit the diffusion of oxygen to a certain level. In this regard, if we can reduce the pore diameter of the separator membrane, then we can reduce the gas crossover, which leads to reduced oxygen diffusion into the cathodic chamber and increases the efficiency of the electrolyzer [35,36].

To overcome the issues related to gas crossover, significant research on alkaline anion exchange membranes (AEMs) was carried out, but these membranes exhibited lower stability as compared to the acidic type of membranes [37,38]. Thus, a lot of notable work was done to decrease the ohmic losses by manufacturing much improved and advanced separator membranes and AEMs. 

Commercial AWEs use porous diaphragm materials as separators because of their high durability. Currently, Agfa’s Zirfon^®^ is used commercially as the diaphragm, Zirfon^®^ is a combination of polysulfone (PSU) and ZrO_2_ nanoparticles. Zirconia, being hydrophilic, provides excellent wettability and stiffness to the separator, while polysulfone is a binder and imparts flexibility. Zirfon^®^ exhibits excellent stability in an aqueous KOH solution. Previously, asbestos was used as a separator membrane for AWE; however, due to its toxic nature, it was replaced. Additionally, Zirfon^®^ exhibits superior properties as compared to asbestos [39,40,41,42,43]. Porous Zirfon^®,^ having a big pore size (up to ~130 nm), promotes the transport of gas-containing electrolyte across the separator, resulting in an increased electrical conductivity as well as gas crossover [44]. This increased gas crossover resulted in a significant reduction in the dynamic range of the electrolyzer [45,46]. Gas crossover can be reduced by increasing the thickness of the porous separator membrane; however, it will also increase the ohmic voltage drop in the electrolyte, resulting in lower energy efficiency [47]. For that reason, the need of the hour is to develop advanced separator membranes having excellent ionic conductivity besides the reduced gas crossover for highly efficient AWEs. 

Several studies have been conducted to manufacture a film with improved ionic conductivity while controlling the pore size to be small. In recent times, a lot of work has been done on thermoplastic nature-based polymer separators owing to the excellent stability and better handling properties of this type of polymer. However, owing to the hydrophobic behavior of these polymers, inorganic hydrophilic filler materials are added to enhance the surface properties of the separator. Polysulfone (PSU) is normally applied as a membrane material owing to its high thermal, chemical, and mechanical stabilities [48,49]. Therefore, the use of PSU as matrix material is very favorable to get higher chemical stability under harsh alkaline environments [50]. Many techniques have been developed to improve the surface characteristics of polysulfone such as sulfonation [51], crosslinking [52], or blending [53]. Several inorganic materials such as ZrO_2_ [54,55], CeO_2_ [56], TiO_2_ [57,58], yttria-stabilized zirconia commonly known as (YSZ) [59], and barite mineral (BaSO_4_) have been added as a filler in the polymer matrix [60]. However, these filler materials exhibited a significant reduction in their mechanical stabilities due to agglomeration and improper mixing with the polymer matrix [45,61,62]. In the past, some effort was made to use a Nafion membrane as a separator material for alkaline water electrolyzer. However, the Nafion membrane exhibited poor performance in KOH electrolyte and gave very low water content, high cell resistance, and high cell potential, which reduced the efficiency of the alkaline water electrolyzer. Although the Nafion membrane showed better performance in NaOH as compared to that in KOH, the properties were not good in comparison to commercial Zirfon^®^ and other membranes used for AWE [63,64].

There exists a great influence of inorganic fillers on ionic conductivities. For this reason, a novel approach was adopted in the literature in which silver ion conducting solid polymer electrolyte was incorporated with activated carbon, which demonstrates high ionic conductivities and acts as a potential choice for energy storage devices for instance solid-state capacitor applications [65]. Nanocomposite polymer electrolytes (silica as inorganic filler material incorporated in the polymer matrix) showed a certain effect on the ionic conductivities. The concentration of filler in a polymer matrix is very important for the ionic conductivities, which affect the performance of energy storage devices [66].

Several membranes were used previously for alkaline water electrolysis systems. Polyvinyl-alcohol-based separator membranes exhibited good thermal stability and adequate mechanical strength [57]. Separator membranes containing different mineral fillers such as BaSO_4_ were also employed and gave slightly better resistivity in comparison to asbestos-based membranes [60,67]. Alkali-doped polyvinyl alcohol polybenzimidazole-based membranes were also utilized for AWE. These membranes indicate good thermal and chemical stabilities in aqueous KOH solution; however, they are expensive [68]. Additionally, these separator membranes exhibited reduced performance in comparison to Zirfon^®^. 

In the present work, we managed to synthesize a highly ionic conductive zirconia toughened alumina (ZTA) (with different Al_2_O_3_ to ZrO_2_ ratio)–based porous composite separator membrane for an alkaline electrolyzer via the phase inversion process. The compatibility of the filler material (ZTA) with the polymer was tested. The ionic resistance and gas permeability of the ZTA-based composite separator membranes were observed by using KOH electrolyte. The electrolysis of the prepared separators was conducted by changing the temperature during electrolysis and varying the amount and flow rate of the KOH electrolyte solution. 

## 2. Materials and Methods

### 2.1. Reagents

All materials were obtained from commercially available sources and used as received without any additional treatment or purification. N-Methyl-2-pyrrolidone (NMP, 99.9%) and polyvinylpyrrolidone (PVP) K90 were received from Sigma-Aldrich. The mean average molecular weight of K90 was 360,000. Polysulfone (PSU, Udel) was obtained from Solvay. ZTA nanoparticles having a size of 80 nm were purchased from U.S. Nano Research. A polyphenylene sulfide (PPS) mesh (PPS80PW, PVF Mesh & Screen Technology) was used as a support material. To obtain 20 wt.% KOH solution, 45 wt.% aqueous solutions of KOH (DAEJUNG) were mixed with suitable deionized water generated by a water purification system (Direct-Q, Millipore). The Brunauer–Emmett–Teller (BET) surface area of the ZTA nanoparticles containing 30% zirconia (Z_30_TA_70_) and 5% zirconia (Z_5_TA_95_) was found to be 14 and 9.82 m^2^g^−1^, respectively. The alumina nanoparticles exhibit a BET surface area of 8.87 m^2^g^−1^. The physical properties of the ZTA nanoparticles are listed in Table 1.

### 2.2. Preparation of Separator

ZTA/PSU-based composite separator (ZTA 85 wt.% and polysulfone 15 wt.%) was synthesized by using the film casting technique. First, NMP (which was used as a solvent) 112.19 g and PVP (additive) 4.479 g were placed in a mixing device (RED 150-D, Pendraulik) and mixed for 40 min at 3000 revolutions per minute (RPM) value. After that, PSU 16.76 g and ZTA nanoparticles 95 g were also added to the mixture, and the slurry was mixed continuously at 3000 RPM to get a homogeneous suspension having the required viscosity. The mixing was done at 40 °C. After preparing a homogeneous and stable suspension, this suspension was then placed on a quartz plate, and casting was done by applying a doctor blade at 40 °C. PPS mesh was added during the casting as supporting material. By using a doctor blade of different heights w.r.t the quartz plate, the separator’s thickness was altered. The prepared sample was then placed in an oven for drying at 80 °C for 15 min. After that, it was introduced in the coagulation bath for extraction. The extracted membrane sample was then stored in deionized water.

### 2.3. Electrode Preparation

Raney nickel and nickel–iron (Ni-Fe) LDH was employed on the cathode and anode side, respectively, as a catalyst. Raney-type materials are the alloys of electrocatalytically active metals such as Ni, Co, and Cu, with readily leached metals such as zinc and aluminum. Active metals can leach out easily in alkaline solutions. These materials are mostly used to enhance the real surface area; however, the surfaces belonging to the rough structures can also exhibit high electrocatalytic activity. Raney nickel, used in the present study, only contains nickel. Nickel foams (NI003852, having a thickness of around 160 μm and porosity of 110 PPI) were acquired from Goodfellow Corp. Pre-treatment of Ni foams was done for 5 min in 20 wt.% NaOH solution and 80 °C temperature to eliminate the surface oxides and impurities and drenched in 18 wt.% HCl solution at room temperature for a few minutes. After the sputtering of aluminum on nickel-based foam, aluminum leaching was done to synthesize a Raney-nickel-type cathodic catalyst. With the help of the physical vapor deposition (PVD) process, aluminum was introduced onto a porous nickel foam (~1.6 mm thickness) by using DC magnetron type sputtering mode at 300. The aluminum coating exhibited a thickness of around 10 μm. It was then heated for 150 min at a temperature of 610 °C. After the successful heating of the aluminum-coated Ni foam, it was placed in 30 wt.% potassium hydroxide (KOH) solution for 24 h, and 80 °C temperature was selected for the selective leaching of aluminum in gamma phase Ni-Al alloys. An in situ pH-controlled growth method was used to fabricate the Ni-Fe double-layered hydroxide (LDH) widely used as the anode. A cleaned iron foam was added into the already prepared nickel sulfate solution and continuous oxygen sparging was carried out at 50 °C and for 7 h into the solution to control the pH level. No external voltage was used during this procedure. The prepared sample was taken out after 7 h and washed with DI water followed by cleaning with ethanol and after that dried in a vacuum desiccator. An Ni:Fe ratio of 1:1 was observed in the prepared LDH type anode, and a fine NI-Fe LDH structure was obtained having CO_3_^−2^ intercalated anions.

### 2.4. Characterization

High-magnification scanning electron microscopy (SEM, S-4800, HITACHI, Tokyo, Japan) was utilized to characterize the morphology of the prepared ZTA separator membranes. To further investigate the existence of microscopic particles in the separator membranes, mapping analysis was carried out by energy dispersive spectroscopy (EDS, Ultim^®^ Max, Oxford Instruments, Abingdon, UK). X-ray diffraction (XRD, Rigaku/D/max-2000 Ultima, Tokyo, Japan) was performed to confirm the crystal structure of the separator membrane and the commercial Zirfon^®^ separators. The physical characteristics such as porosity, pore volume, and the average pore diameter of the prepared separator membranes were studied by mercury porosimeter analysis (Micromeritics, Auto Pore 9520, Norcross, GA, USA).

To calculate the ionic conductivity of the prepared ZTA separator membranes, electrochemical impedance spectroscopy (EIS) (Biologic science Instruments, SP240) analysis was performed by using an H-shaped cell (VB8-S EC frontier). The prepared separator membrane was inserted in the H-type cell having an active area of 3.14 cm^2^. The experiment was carried out by pouring 30 wt.% aqueous KOH solution in the H-type cell, and the measurements were recorded by an AC-type amplitude, which corresponds to 10% of utilized current density in 10 kHz–0.1 Hz frequency range, and ionic conductivities of prepared separator membranes were measured at a frequency of 2 kHz.

To calculate the hydrogen gas permeability, a 24 cm^2^ size separator sample was cut and placed between the two halves of the cell. Both sides of the cell were packed with 30 wt.% KOH. A mass flow meter, as well as a back-pressure-based regulator, was utilized to control the differential pressure from 1.1 to 1.5 bar. The side with the higher pressure was referred to as the cathode and the other side having low pressure was called the anode. The mass of KOH electrolyte passed across the separator membrane was measured. The hydrogen permeability $\varepsilon_{H_{2}}^{\mathrm{Darcy}}$ (in mol cm^−1^ s^−1^ bar^−1^) induced by the difference of pressure is denoted as follows:(1)$$\varepsilon_{H_{2}}^{\mathrm{Darcy}}=\frac{K}{\eta }S_{H_{2}}P_{H_{2}}^{\mathrm{cat}}$$
where K represents the permeability of the electrolyte (cm^2^), $S_{H_{2}}$ indicates the hydrogen gas solubility in the electrolyte (mol m^–3^ bar^–1^), $P_{H_{2}}^{\mathrm{cat}}$ represents the partial H_2_ pressure at the cathodic side (bar), and the electrolyte’s viscosity is denoted by η (bar s). In accordance with the Darcy law, the molar permeation flux density ($\Phi_{H_{2}}^{\mathrm{Darcy}}$ in mol s^–1^ bar cm^–2^) of hydrogen gas across the separator membrane induced by the difference of pressure was calculated using the following: (2)$$\Phi_{H_{2}}^{\mathrm{Darcy}}=-\varepsilon_{H_{2}}^{\mathrm{Darcy}}\frac{\Delta p}{d}$$
where Δp represents the absolute differential pressure (bar) between the cathode and the anode and  d indicates the separator thickness.

The stability of Z_30_TA_70_ was tested by putting the separator in 35 wt.% KOH at 100 °C for two and four weeks. XRD and Brunauer–Emmett–Teller (BET) analysis were performed after the stability test to investigate any changes in the crystal structure and physical properties of Z_30_TA_70_, respectively.

### 2.5. Electrolysis Tests

The electrochemical characterization was conducted in a conventional three-electrode configuration in 29.9 wt.% KOH solution at 333 K with a special sample holder (as described elsewhere [69]). For the experiments, a two-compartment cell was used, and the half cells were separated by the diaphragm (Zirfon^®^ Perl 500). A reversible hydrogen electrode was used as a reference electrode (HydroFlex from Gaskatel). For the OER test, the potential versus the RHE was calculated to 1.202 V, using the equation as given elsewhere [70]. Electrochemical experiments such as (cyclic voltammetry (CV), galvanostatic experiment (GS), Tafel plot) were carried out to examine the HER activity of the electrodes. A standard test protocol was used, which includes the steps as listed below:GS: determining overpotential (*η*_500_) for HER/OER at ± 500 mA/cm^2^ after 5 h.Tafel plot: potential-current density behavior.Pretreatment CV: removing adsorbed and absorbed H-species.CV: determining C_dl_ at OCP after HER.

Steady-state polarization curves were also recorded to display the Tafel plot. The determination of the C_dl_ from the CV data and the reason for the pretreatment CV are given in a previous publication [69].

### 2.6. In Situ Cell Tests

The electrolysis of prepared separator membranes was carried out in a single zero-gap-based configuration cell with a separator membrane consisting of 85 wt.% ZTA and 15 wt.% PSU, taking Zirfon^®^ as the reference material. The cell had an active area of around 34.56 cm^2^. The cell contained the current collectors, bipolar plates, and nickel-based porous transport layers (PTLs). The performance of the cells with the prepared membrane was analyzed at 80 °C, and the KOH electrolyte was circulated at different flow rates. The catalyst applied at the cathode side was Raney nickel (composed of nickel only) and at the anode side Nickel–Iron (Ni-Fe) LDH was used. With both catalysts, nickel foam was also additionally used as the catalyst. The voltage of the cell was recorded at current densities ranging from 0 to 2000 mA cm^–2^ with a potentiostat (Advanced Power System Keysight N7970A, Agilent Technologies). Galvanostatic electrochemical impedance spectroscopy (GEIS) was used for measuring at 200 mA cm^−2^ of current density within the range between 0.1 MHz and 1 Hz.

## 3. Results

### 3.1. Characterization of Prepared Separator Membranes

Figure 1 shows the X-ray diffraction analyses of commercial Zirfon^®^ and the prepared separator membranes, which were carried out to confirm the crystal structure. The presence of the monoclinic zirconia phase in Zirfon^®^ was confirmed by the standard card of JCPDS #37-1484 [71]. The alumina separator exhibited a hexagonal structure by JCPDS #46-1212 [72]. In the Z_5_TA_95_ and Z_30_TA_70_ separator membranes (zirconia nanoparticles in alumina matrix), the presence of large hexagonal phase and smaller monoclinic phases was confirmed by JCPDS #46-1212 and JCPDS #37-1484, respectively.

Scanning electron microscopy is utilized for the morphological analysis of ZTA separator membranes and Zirfon^®^. Figure 2 represents the cross-sectional view of the ZTA separator membrane and Zirfon^®^.

The reference Zirfon^®^ was topped by a few-micrometer-thick rich polysulfone (PSU) layer (Figure 2e). A random distribution of zirconia nanoparticles and PSU composite under the rich polysulfone (PSU) layer was visible in the case of Zirfon^®^ (Figure 2e). No such layer was observed in the alumina separator membrane (Figure 2b,f). Additionally, such a thick layer was also not observed on the top of the Z_5_TA_95_ (Figure 2c,g) and Z_30_TA_70_ separator membranes (Figure 2d,h). The alumina separator membrane exhibits that the particle shape of the alumina matrix is mainly spherical (Figure 2b,f). The Z_5_TA_95_ membranes, which have a very low content of zirconia (only 5%), also exhibit the presence of a spherical matrix in Figure 2c,g, with the increase of zirconia content in the alumina matrix; as in Z_30_TA_70_, the original spherical shape of particles disappears in Figure 2d,h. ZTA nanoparticles are properly mixed with polysulfone, i.e., distributed uniformly and display good compatibility with polymer as shown in Figure 2c,h. SEM-EDS analysis confirmed the presence of Al in the alumina separator membrane (Figure 2j). It can be confirmed, that Al and Zr are present in the ZTA separator membrane (Figure 2k,l). Mapping also exhibits that zirconia particles are present at the boundary of the alumina matrix and are uniformly distributed with alumina (Figure 2k,l).

Figure 3a shows the mercury pore size distribution of ZTA separator membranes and reference Zirfon^®^. Separator membranes are largely comprised of nanometer-size interconnected ZTA pores. Small pores of 10 to 50 nm represent the pores between the ZTA nanoparticles and relatively larger, macro-sized pores of are correlated with the interconnected polysulfone pore network. Zirfon^®^ gave an initial broad peak within the range of 10–50 nm and a second narrow and large peak having strong intensity at around 1000 nm pore size. However, the ZTA-based separator membranes displayed a reduction in the micro-sized type pores with a decrease in the intensity as well. The first peak intensity of the alumina and Z_5_TA_9*5*_ separator membranes (at around 70 nm) was greater than that of the Z_30_TA_70_ separator membrane, which indicates that alumina and Z_5_TA_9*5*_ have a smaller average pore size as compared to Z_30_TA_70._ However, the prepared separators exhibited a wide distribution of pores from 50 to 100 nm and demonstrated higher-intensity values as compared to Zirfon^®^. The Z_30_TA_70_ exhibits the second peak of large intensity at greater than 1000 nm. The physical characteristics of the separator membranes and Zirfon^®^ are summarized in Table 2. The largest pore size (~125.10 nm) was observed in the Z_30_TA_70_ separator membrane. The reference Zirfon^®^ represented the average pore size value (~99 nm), which was lower than that of Z_30_TA_70._ The alumina separator membrane had the smallest average pore size value (~79.63 nm). It was noted that the ZTA separator membranes exhibited a wider pore distribution from 50 to 100 nm, which is related to the peak shift onto the left side toward the smaller pore diameters in comparison with the reference Zirfon^®^ (Figure 3a). The alumina separator membrane having a spherical pore shape exhibited a large pore area (16.39 m^2^ g^−1^) (Table 2), in comparison to the Z_30_TA_70_ membranes, which showed a 13.65 m^2^ g^−1^ pore area. Z_30_TA_70_ with a large average pore diameter exhibited uniform pore distribution throughout the structure as shown in Figure 2h.

The ohmic resistance of the separator membrane and Zirfon^®^ is shown in Figure 3b. KOH is mostly used as an electrolyte in the case of an alkaline electrolyzer. Ionic conductivity is supplied by an aqueous alkaline solution that penetrates through the pores of the separator membrane. The ionic conductivity of the separator membrane is directly linked to their ionic resistance. The ionic conductivity of the KOH electrolyte used in this study was 0.4468 S/cm. Zirfon^®^ (prepared with 40 nm zirconia nanoparticles) exhibited area resistance around 0.3 Ω·cm^2^. The ohmic resistance of the prepared separator membranes was lower than that of Zirfon^®^. The separator membrane containing only alumina particles (Alumina: ZrO_2_; 100:0) showed an area resistance of around 0.172 Ω·cm^2^, which is lower than that of commercial Zirfon^®^. The area resistance decreased further when we used the ZTA separator membrane. The Z_5_TA_95_ exhibited an area resistance of 0.159 Ω·cm^2^. It reduced further to 0.15 Ω·cm^2^ as the amount of zirconia in alumina increased to 30 wt.%, i.e., Z_30_TA_70_ (Figure 3b).

The ionic conductivity is supplied by the KOH electrolyte solution, which permeates through the pores of the porous separator membrane. The porous separator membrane separates the evolved gases [73]. The KOH electrolyte solution can easily pass across the separator membrane due to the induced differential pressure between the two electrode sides, i.e., the cathodic and anodic sides. With the increase in the amount of porosity, the permeability values also increased. The highly porous separator membranes exhibited low ionic resistance; however, they demonstrated high permeability. Therefore, there is a compromise between the two most important properties of separator membranes, i.e., ohmic resistance and H_2_ gas permeability. The reference Zirfon^®^ contained micropores having strong intensity values as shown in Figure 3a. The dissolved hydrogen gas in the electrolyte can pass through the micropores of porous Zirfon^®^ due to the induced differential pressure [74], which gives rise to a low partial load capability of AWEs. On the other hand, the polymer-based electrolyte membrane in the case of a PEM electrolyzer normally consists of nanopores of a few nanometers around (2–5 nm), due to which they showed low crossover of gas induced by the differential pressure. To reduce the gas crossover induced by the difference of pressure, porous separator membranes with small pores can be developed.

Zirfon^®^ exhibited high H_2_ permeability in the order of 26.8 × 10^−12^ mol cm^−1^ s^−1^ bar^−1^, as seen in Figure 3c. However, in comparison to the reference Zirfon^®^, the prepared ZTA separator membranes showed less H_2_ permeability. The 100% Al_2_O_3_ separator membrane exhibited H_2_ permeability of 14.2 × 10^−12^ mol cm^−1^ s^−1^ bar^−1^. With the increase in the amount of zirconia in alumina, permeability decreases. The ZTA 5% separator membrane, Z_5_TA_95_, showed H_2_ permeability of around 11.2 × 10^−12^ mol cm^−1^ sec^−1^ bar^−1^; however, Z_30_TA_70_ showed a further reduction in H_2_ permeability, having 10.7 × 10^−12^ mol cm^−1^ s^−1^ bar^−1^ in Figure 3c.

#### Characterization of the Anode and Cathode Material

The cross-section analysis of the Raney-Ni cathode is given in Figure 4 in the as-leached state before the electrochemical tests. Due to the selective coating onto only one side of the substrate, the Raney-Ni coating could only be observed at one side (marked in Figure 4b with yellow arrows). The layer thickness was around 5–10 µm. The SEM-EDX analysis (Figure 4a) revealed that the Al coating has transformed into a Ni-Al alloy. The Al content was still quite high after the leaching process (see also Figure 4d). No delamination of the Raney-Ni layer was detected.

Contrary to the Raney-Ni electrode, the Ni-Fe preparation process leads to a homogeneously coated Fe-foam with an onion-like Ni-Fe-S layer (see Figure 5). The layer thickness was around 10–20 µm. Besides the outer sides, also at the inner sides of the foams, the Ni-Fe-S layer could be seen. The elemental mapping revealed that the coating consisted of two separated components: Ni-S and Fe-O-S. The Ni-S appeared to be the matrix and Fe-O-S was embedded in the matrix. The sulfur content in the sample was introduced by the electroplating process. High porosity was observed. 

The samples were electrochemically investigated using the test protocol given in Section 2.5. The results in Figure 6 reveal that the Raney-Ni-coated foam exhibited an overpotential after 5 h of *η*_500_ = 174 mV. A slight increase of potential was detectable (~+5 mV/h). 

The high electrocatalytic activity toward the HER was also reflected by the high surface area (C_dl_ = 0.040 F/cm^2^, extracted from Figure 7a) and the low Tafel slope of 79.9 mV/dec (in the high current density region). The high C_dl_ value indicates that the high activity was a result of the high surface area (extrinsic activity).

The Ni-Fe material in turn showed a much higher absolute overpotential (here for the OER *η*_500_ = 274 mV); however, a decrease of the overpotential over time (−1.6 mV/h) was seen. As can be seen in Figure 7b, the CVs for the anodic region are not parallel to the x-axis. Thus, the determination of the C_dl_ was subject to uncertainty. However, the average C_dl_ of 0.053 F/cm^2^ was within the range of the anodic C_dl_ (0.061 F/cm^2^) and the cathodic C_dl_ (0.044 F/cm^2^). The onion-shell structure seems to contribute to the formation of a high electrocatalytic surface. The Tafel plot exhibited a Tafel slope of 90.6 mV/dec in the high current density region. It can be concluded from the electrochemical analysis that the Ni-Fe anode material activity is mainly limited by low intrinsic activity. This cannot be overcompensated by the high surface area. 

### 3.2. Cell Test of the Separator Membrane

The electrolysis of ZTA separator membranes was performed by using different metallic catalysts and nickel foams as the porous transport layer. The polarization properties of cells combined with different separator membranes can be seen in Figure 8a.

The electrolysis test was conducted at 80 °C temperature by using 30 wt.% aqueous solutions of KOH and electrodes (Raney nickel on cathode and Ni-Fe LDH at anode). The cell carrying the Z_30_TA_70_ separator membrane exhibited a much-improved performance of 2.1 V at 2 A/cm^2^ in comparison to the reference Zirfon^®^, which showed 2.41 V at 2 A/cm^2^. This makes Z_30_TA_70_ more efficient as compared to Zirfon^®^. The other separator membranes also show improved performance as compared to Zirfon^®^ as shown in Figure 8a. The ohmic resistance of the Z_30_TA_70_ separator membrane was analyzed by using the Nyquist plot. Figure 8b shows the Nyquist plot of impedance spectra at 1 A/cm^2^. The equivalent circle largely consists of ohmic resistance, activation resistance, and mass-transport resistance. The high-frequency resistance (HFR) is demonstrated by the intercepts of the Nyquist plot with the x-axis at higher frequency (left portion on Nyquist plot), representing the ohmic resistance of the cell. The ohmic resistance is the resistance caused by the flow of current through the cell, which is mainly due to the contribution from the membrane. The area resistance measure ex situ in Figure 3b matched well with the in situ internal resistance of Figure 8b. The middle portion of the equivalent circuit model exhibited the activation loss. The activation loss results from the kinetics of electrodes. Only one semicircle from the high to the middle frequency with a similar size was observed for all the membrane separators, which is attributed to the use of the same electrodes. The mass-transport limitation and resistance were not observed in this study.

The performance of the improved Z_30_TA_70_ separator membrane was analyzed by using different electrolyte concentrations (10, 20, and 30 wt.%), by varying the temperature of electrolysis from 50 to 80 °C and by using various KOH flow rates. Figure 9b shows that the Z_30_TA_70_ membrane exhibited improved performance in the 30 wt.% KOH solution. It exhibited 2.1 V at 2 A/cm^2^. This enhanced performance was due to the uniform distribution of Z_30_TA_70_ in polymer matrix (Figure 2h), and the low ohmic resistance (0.15 Ω·cm^2^) of Z_30_TA_70_ (Figure 3b). However, Zirfon^®^ showed the same current density at a higher voltage of 2.4 V (Figure 9a). 

The low performance of Zirfon^®^ in comparison to Z_30_TA_70_ was due to the random distribution of zirconia nanoparticles in polymer matrix (Figure 2e), and the higher ohmic resistance 0.3 Ω·cm^2^. Z_30_TA_70_ exhibited reduced performance in 20 wt.% and 10 wt.% KOH (Figure 9b); however, it was still much better than Zirfon^®^. The effect of KOH concentration on cell voltage at different current densities was also analyzed (Figure 9c,d). There was no change in the performance of Z_30_TA_70_ and Zirfon^®^ at various KOH concentrations at a low current density of 0.1 A/cm^2^ (Figure 9c,d). On the contrary, at a higher current density of 2 A/cm^2^, the performances of both Z_30_TA_70_ and Zirfon^®^ were enhanced by increasing the KOH concentration from 10 to 30 wt.%. Zirfon^®^ exhibited a high cell voltage of 3.6 V at a high current density of 2 A/cm^2^ owing to the use of diluted aqueous KOH solution of 10 wt.%. It is well known that ionic conductivity values decrease with lower KOH concentration, which implies that the internal resistance of Zirfon^®^ is relatively higher than that of our prepared membranes. 

The polarization properties of Z_30_TA_70_ were also studied by varying the temperature during the test. The Z_30_TA_70_ exhibited excellent performance at 80 °C; however, performance was reduced by lowering the temperature from 80 to 50 °C due to the increase in the viscosity of KOH solution at low temperature. It indicates that 80 °C is the optimum electrolysis temperature for AWEs (Figure 10b). 

Zirfon^®^ also showed reduced performance at lower temperatures and exhibited improved performance at 80 °C (Figure 10a); still, it showed lower performance than Z_30_TA_70_ (Figure 10a,b). The effect of temperature on cell voltage at different current densities was also analyzed (Figure 10c,d). At a higher current density of 2 A/cm^2^, the performances of both Z_30_TA_70_ and Zirfon^®^ were enhanced by increasing the temperature to 80 °C (Figure 10c,d). The reason is that, at high temperatures, the viscosity of the KOH solution decreases, which increases the flow of KOH.

The polarization characteristics of Z_30_TA_70_ were also examined by varying the KOH circulation rate. By increasing the circulation rate of KOH, the performance of Z_30_TA_70_ also improved. Z_30_TA_70_ demonstrated 2 A/cm^2^ at 2.45 V at 50 mL/min KOH flow rate. On increasing the flow rate to 800 mL/min, Z_30_TA_70_ exhibited improved performance of 2 A/cm^2^ at 2.1 V due to an increase in KOH flow through the separator membrane (Figure 11b). For comparison, Zirfon^®^ showed reduced performance as compared to Z_30_TA_70_ at different flow rates due to its higher ohmic resistance and gas permeability (Figure 11a). The effect of the circulation rate of KOH on cell voltage at different current densities was also analyzed (Figure 11c,d). At a higher current density of 2 A/cm^2^, the performances of both Z_30_TA_70_ and Zirfon^®^ were enhanced by increasing the KOH circulation rate. This was mainly due to the rapid flow of KOH through the separator membrane. However, Z_30_TA_70_ exhibited much better performance than Zirfon^®^ (Figure 11c,d).

To investigate the performance of Z_30_TA_70_, a stability test was performed in harsh conditions. Z_30_TA_70_ was immersed in 35 wt.% aqueous solutions of KOH at 100 °C for four weeks. The stability was checked after two weeks and four weeks by conducting XRD and BET analysis, as shown in Figure 12.

No significant change in the XRD peak was observed after the stability test (Figure 12a). The physical properties of Z_30_TA_70_ after the stability test are given in Table 3. There was no change in BET surface area after the stability test. A slight increase in pore size was observed after the stability test, which also led to a small increase in the volume of the pore (Figure 12b). However, this change was very negligible and did not affect the overall properties of Z_30_TA_70._ The stability test indicated that Z_30_TA_70_ had very little degradation in harsh conditions.

## 4. Conclusions

The development of a highly ionic conductive and durable porous separator membrane is vital for enhancing the performance of an alkaline water electrolyzer. This paper suggested a Z_30_TA_70_-based separator membrane that exhibits high ionic conductivity and low gas permeability in comparison to state-of-the-art Zirfon^®^. The Z_30_TA_70_ separator membrane, having a low thickness of 430 μ m, exhibited low ohmic resistance 0.15 Ω·cm^2^ and lower hydrogen gas permeability 10.7 × 10^−12^ mol cm^−1^ s^−1^ bar^−1^. Raney nickel and Ni-Fe LDH electrodes were prepared successfully. Electrochemical investigations were carried out in which Raney-Ni-coated foam exhibited an overpotential after 5 h of *η*_500_ = 174 mV and exhibited high electrocatalytic activity toward the HER owing to its high surface area. On the other hand, the Ni-Fe LDH showed a much higher absolute overpotential (here, for the OER *η*_500_ = 274 mV). It can be concluded from the electrochemical analysis that the Ni-Fe anode material activity is mainly limited by low intrinsic activity. This cannot be overcompensated by the high surface area. The electrolytic cell containing Z_30_TA_70_ separator membrane having Raney nickel as cathode and Ni-Fe LDH as anode and 30 wt.% KOH electrolyte demonstrated 2 A/cm^2^ at 2.1 V. The performance of this improved Z_30_TA_70_ separator membrane was also analyzed by varying the KOH electrolyte concentration, the KOH flow rate, and the electrolysis temperature. The results demonstrate that the Z_30_TA_70_ separator membrane gives an improved performance as compared to the reference Zirfon^®^. The results presented in this paper make the Z_30_TA_70_ separator membrane a suitable material for highly advanced and efficient alkaline water electrolyzer systems.

## Figures and Tables

**Figure 1 polymers-14-01173-f001:**
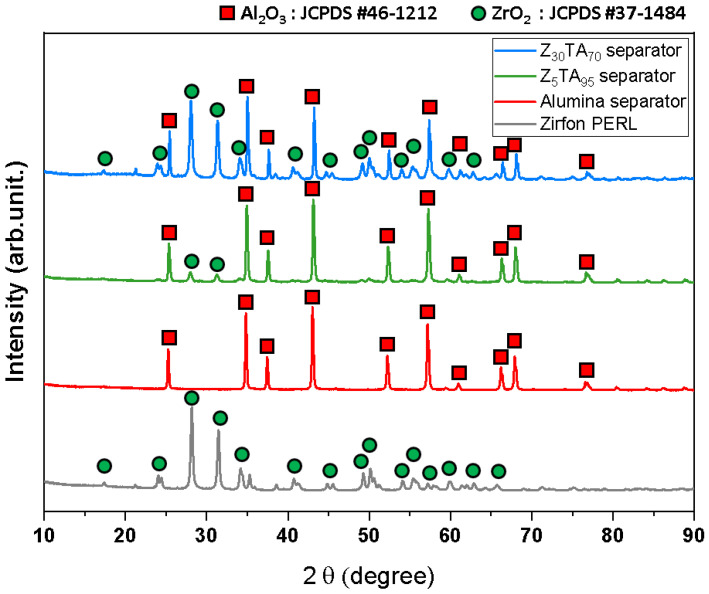
XRD patterns of commercial Zirfon^®^ and prepared separator membranes.

**Figure 2 polymers-14-01173-f002:**
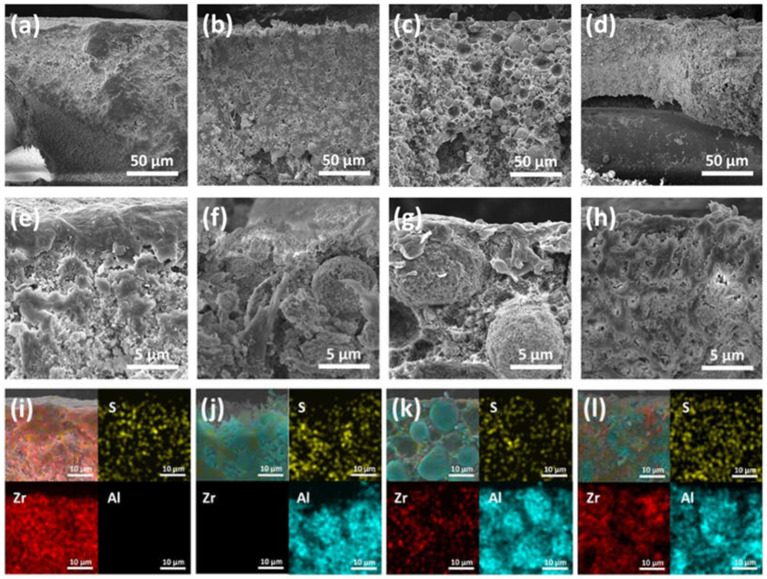
Cross-sectional SEM images, low-magnification (**a**) Zirfon^®^, (**b**) alumina separator membrane, (**c**) Z_5_TA_95_ separator membrane, and (**d**) Z_30_TA_70_ separator membrane and higher-magnification (**e**) Zirfon^®^, (**f**) alumina separator membrane, (**g**) Z_5_TA_95_ separator membrane, and (**h**) Z_30_TA_70_ separator membrane. SEM-EDS elemental mapping analysis of (**i**) Zirfon^®^, (**j**) alumina separator membrane, (**k**) Z_5_TA_95_ separator membrane, and (**l**) Z_30_TA_70_ separator membrane.

**Figure 3 polymers-14-01173-f003:**
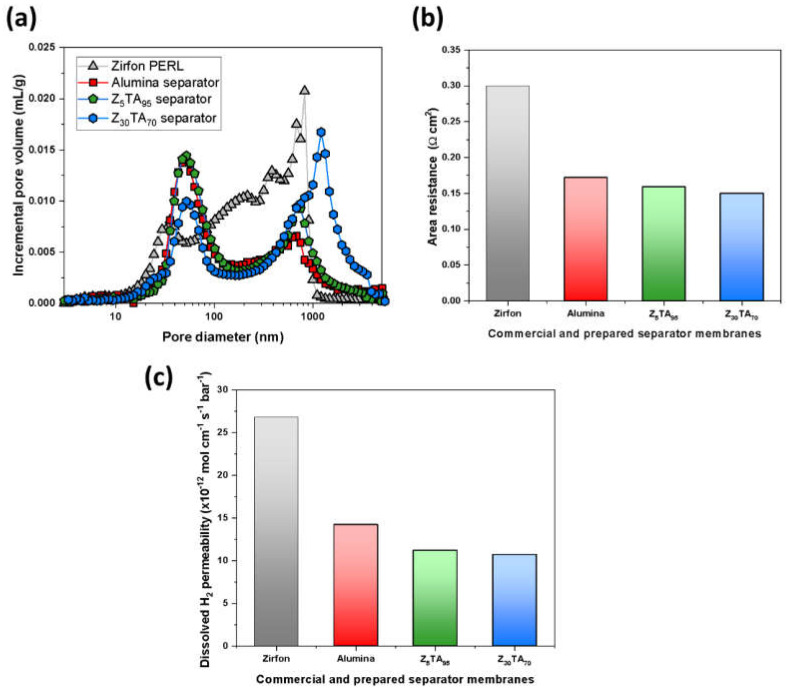
(**a**) Mercury incremental intrusion volume curves of Zirfon^®^ and the prepared separator membranes. (**b**) Area resistance of Zirfon^®^ and the prepared separator membranes. (**c**) Permeability of Zirfon^®^ and the prepared separator membranes.

**Figure 4 polymers-14-01173-f004:**
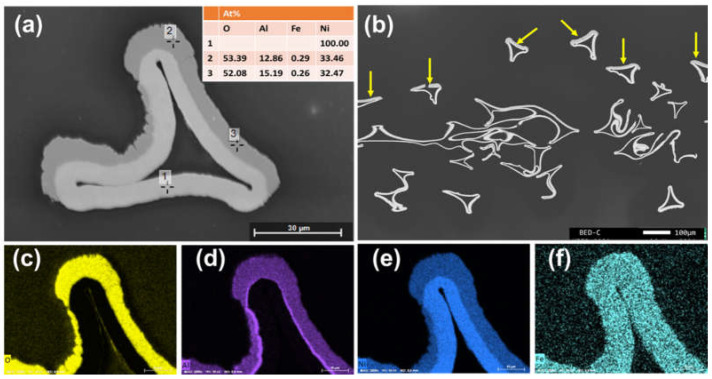
Cross-sectional SEM images of Raney-Ni electrode in the as-leached state (**a**,**b**) and elemental mapping images for (**c**) oxygen, (**d**) aluminum, (**e**) nickel, and (**f**) iron.

**Figure 5 polymers-14-01173-f005:**
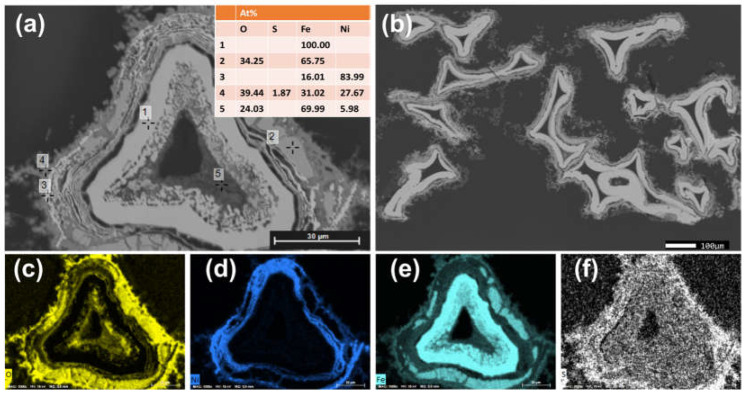
Cross-sectional SEM images of Ni-Fe electrode in the as-prepared state (**a**,**b**) and elemental mapping images for (**c**) oxygen, (**d**) nickel, (**e**) iron, and (**f**) sulfur.

**Figure 6 polymers-14-01173-f006:**
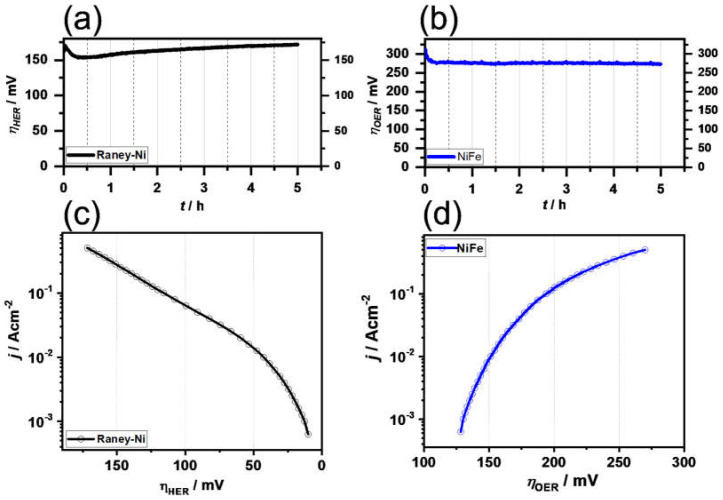
Galvanostatic curves for (**a**) Raney nickel and (**b**) Ni-Fe electrode, and polarization curves for an electrochemical cell using (**c**) Raney nickel and (**d**) Ni-Fe electrode, in 30 wt.% KOH solution at 333 K.

**Figure 7 polymers-14-01173-f007:**
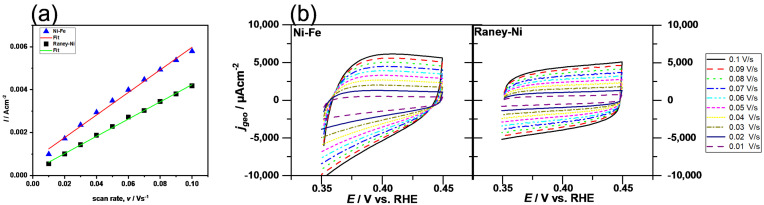
Galvanostatic curves for (**a**) plot of the average current density at 0.4 V (vs RHE) vs the scan rate (extracted from (**b**)) and CVs for Ni-Fe and Raney Ni electrodes at different scan rates.

**Figure 8 polymers-14-01173-f008:**
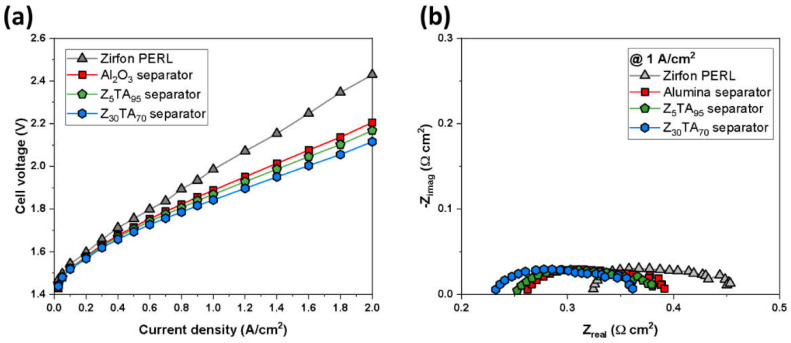
**(a)** Polarization curves for an electrochemical cell using Raney nickel as the cathodic and Ni-Fe LDH as the anodic electrode, respectively, in 30 wt.% aqueous solutions of KOH at 80 °C temperature. (**b**) Nyquist plot of impedance spectra at 1 A/cm^2^.

**Figure 9 polymers-14-01173-f009:**
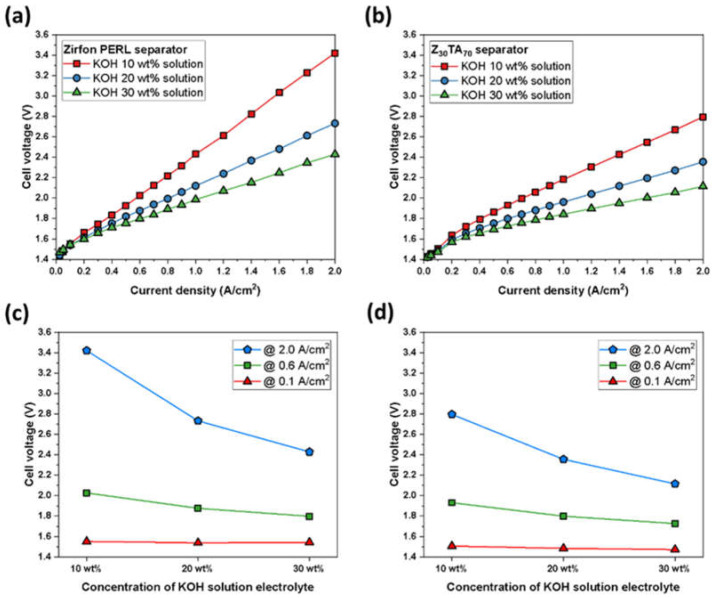
Polarization curves for an electrochemical cell using Raney nickel as the cathodic and Ni-Fe LDH as anodic electrode, respectively, in various KOH concentrations (**a**) Zirfon^®^ and (**b**) Z**_30_**TA_70_ separator membrane. Effect of KOH concentration on cell voltage at different current densities, (**c**) Zirfon^®^ and (**d**) Z_30_TA_70_ separator membrane.

**Figure 10 polymers-14-01173-f010:**
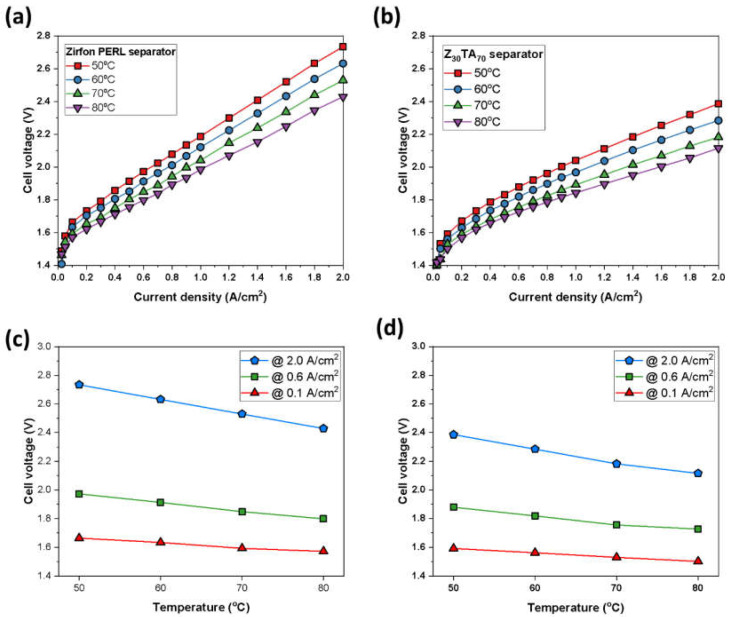
Polarization curves for an electrochemical cell using Raney nickel as the cathodic and Ni-Fe LDH as anodic electrode, respectively, at different temperatures (**a**) Zirfon^®^ and (**b**) Z_30_TA**_70_** separator membrane. Effect of temperature on cell voltage at different current densities (**c**) Zirfon^®^ and (**d**) Z_30_TA_70_ separator membrane.

**Figure 11 polymers-14-01173-f011:**
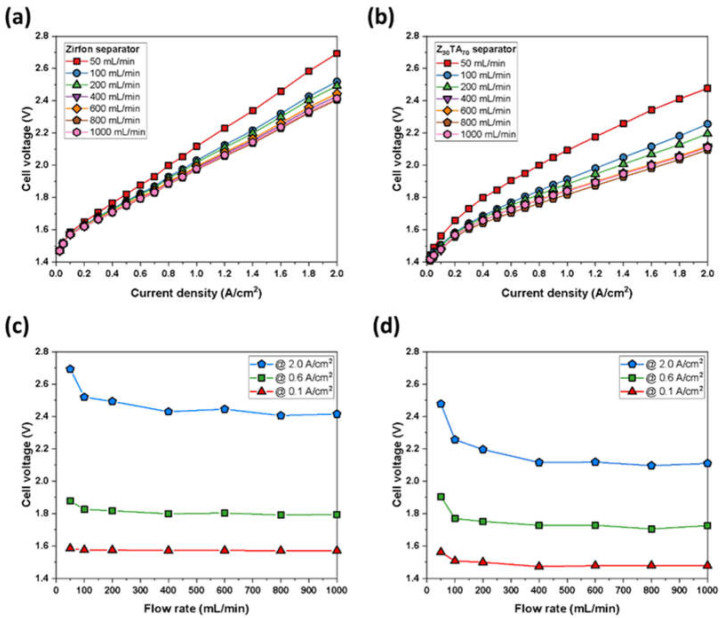
Polarization curves for an electrochemical cell using Raney nickel as the cathodic and Ni-Fe LDH as anodic electrode, respectively, at various KOH flow rates (**a**) Zirfon^®^ and (**b**) Z_30_TA_70_ separator membrane. Effect of KOH flow rate on cell voltage at different current densities (**c**) Zirfon^®^ and (**d**) Z_30_TA_70_ separator membrane.

**Figure 12 polymers-14-01173-f012:**
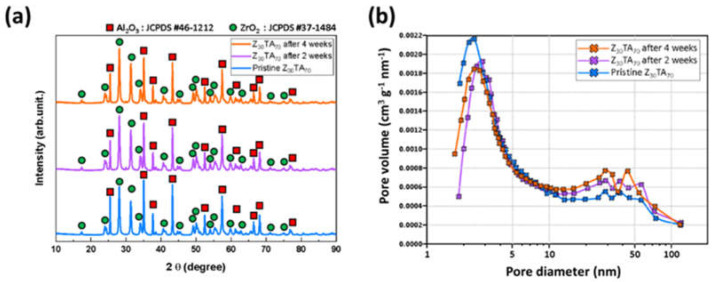
The stability of Z_30_TA_70_: (**a**) XRD of Z_30_TA_70_ after two and four weeks (**b**) BET of Z_30_TA_70_ after two and four weeks.

**Table 1 polymers-14-01173-t001:** The physical attributes of ZTA-based nanoparticles.

Samples	BET Surface Area (m^2^ g^−1^)	Pore Volume (cm^3^ g^−1^)
Al_2_O_3_:ZrO_2_ (70:30) Z_30_TA_70_	14	0.060
Al_2_O_3_:ZrO_2_ (95:5) Z_5_TA_95_	9.82	0.027
Al_2_O_3_:ZrO_2_ (100:0)	8.87	0.025

**Table 2 polymers-14-01173-t002:** The physical properties of Zirfon^®^ and the prepared separator membranes.

	Unit	Z_30_TA_70_	Z_5_TA_95_	Al_2_O_3_:ZrO_2_ (100:0)	Zirfon^®^
Total pore area	[m^2^ g^−1^]	13.65	15.85	16.39	19.26
Average pore diameter	[nm]	125.10	81.68	79.63	99.06
Bulk density	[g/mL]	1.26	1.42	1.40	1.10
Apparent density	[g/mL]	2.75	2.64	2.58	2.34
Porosity	[%]	54.06	46.09	45.78	52.82
Thickness	μ m	430 ± 5	430 ± 15	460 ± 5	470 ± 10

**Table 3 polymers-14-01173-t003:** The physical properties of Z_30_TA_70_ after stability test.

Sample	BET Surface Area (m^2^ g^−1^)	Pore Volume (cm^3^ g^−1^)	Pore Size (nm)
Z_30_TA_70_	14	0.060	17
Z_30_TA_70_ after 2 weeks	14	0.065	19
Z_30_TA_70_ after 4 weeks	14	0.070	20

## Data Availability

Data that support the findings of this study are included in the article.

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
