# Peer review of "Zirconia Toughened Alumina-Based Separator Membrane for Advanced Alkaline Water Electrolyzer"

_polymers, 2022, doi:10.3390/polym14061173_

Round 1

Reviewer 1 Report

Hydrogen is an abundant, renewable, and clean energy source and has been considered as a solution to the problems arising from the current unsustainable fossil fuel economy. Recently, the hydrogen economy is gaining the attention of government bodies and major oil companies. Hydrogen can be generated through various methods including thermochemical, electrochemical, and biochemical. Of all these available methods, water electrolysis (via electrochemical) is suitable for the large-scale production of (green) hydrogen. Alkaline electrolysis has become a well-matured technology for hydrogen production up to the megawatt range and constitutes the most extended electrolytic technology at a commercial level worldwide. The aspect for alkaline electrolyzers is the low maximum achievable current density, due to the high ohmic losses across the liquid electrolyte and diaphragm.

The manuscript is well presented with supportive physical and electrochemical validations; it can be considered after some revision.

Following are my specific comments:

  • For KOH liquid electrolytes, the inability to operate at high pressure, which makes for a bulky stack design configuration – how to overcome this issue?
  • The diffusion of oxygen into the cathode chamber reduces the efficiency of the electrolyser – how to overcome this issue?
  • The low gas crossover rate of the solid polymer electrolyte membrane allows for the PEM electrolyser to work under a wide range of power input; how this can be justified for AWE using liquid KOH electrolyte?
  • How about using Nafion membrane? And having ternary mixtures of electrocatalysts?
  • Please provide the composition for Raney nickel; ionic conductivity of the KOH electrolyte.
  • In the last part of the introduction lines 80-82; A range of other inorganic fillers and their influence on ionic conductivity has also been reported in the literature, doi.org/10.1016/j.electacta.2014.06.039; doi.org/10.1021/ie502615w – please include and expand the discussion.
  • In the results and discussion – Text relating Figure 4b (lines 316 -319); Nyquist plot of impedance spectra has not been explained well. Please provide ohmic resistance values and explain the shape of the diameter of the semicircles. Some electrochemical insights on the Nyquist plot (using the background cited in the literature 10.1039/C8NR03824D) will assist the readers to understand the impedance and its role on the mass transport aspects and ohmic resistances.
  • In Fig. 5c how the cell voltage has been as high as 3.6 V while using aqueous solutions? Justify.

Author Response

Response to Reviewer 1 Comments

Points: Hydrogen is an abundant, renewable, and clean energy source and has been considered as a solution to the problems arising from the current unsustainable fossil fuel economy. Recently, the hydrogen economy is gaining the attention of government bodies and major oil companies. Hydrogen can be generated through various methods including thermochemical, electrochemical, and biochemical. Of all these available methods, water electrolysis (via electrochemical) is suitable for the large-scale production of (green) hydrogen. Alkaline electrolysis has become a well-matured technology for hydrogen production up to the megawatt range and constitutes the most extended electrolytic technology at a commercial level worldwide. The aspect for alkaline electrolyzers is the low maximum achievable current density, due to the high ohmic losses across the liquid electrolyte and diaphragm.

The manuscript is well presented with supportive physical and electrochemical validations; it can be considered after some revision.

Response:  We are grateful to the reviewer for the positive remarks.

Point 1: For KOH liquid electrolytes, the inability to operate at high pressure, which makes for a bulky stack design configuration – how to overcome this issue?

Response 1:  We are grateful to the reviewer for the comment. Alkaline water electrolysis is one of the mature technologies having low cost and high production rate. Liquid KOH is used as an electrolyte for ion transfer in the case of the Alkaline water electrolysis system. When the membrane suffered from high gas permeability then it causes high gas crossover issues and made the operation difficult at a high-pressure level and reduces the overall performance. However, it can be overcome by adjusting the differential pressure between cathode and anode carefully at the operational level.

The high-Pressure alkaline water electrolysis plant is now being successfully operated up to high pressure 30 bar. 

The followings are the link.

https://hidrogenoaragon.org/en/productos/high-pressure-alkaline-electrolyzer/

https://cordis.europa.eu/project/id/736351

https://www.sunfire.de/en/news/detail/demo4grid-project-partners-successfully-install-a-3-2-mw-pressurized

To overcome this issue at the material level, we developed a low permeability membrane, having low gas crossover, and enough integrity so that it can use easily in a High-pressure alkaline electrolyzer system. By reducing the gas permeability, we can limit the gas crossover problems which would be able to help to operate at high-pressure conditions with ease.   

[Page 2] AWE is in commercial use for quite a long time and is considered a mature technology [26]. However, they exhibit low operating pressure and limited partial load range, slow response time towards dynamic operations, and most importantly low current densities (below 800 mA/cm2 @ 1.8 V), owing to the higher values of ionic resistance of the separator membrane and the lower kinetics caused by the utilization of non-noble type catalysts [27-29]. These lower densities increased the number of stacks required to obtain enough hydrogen [30-32]. Another reason for the low operating pressure and low partial load range is the non-negligible permeability of dissolved gas in the liquid electrolyte i.e., KOH through the separator membrane [33, 34]. When the membrane suffered from high gas permeability then it causes high gas crossover issues and made the operation difficult at the high-pressure level. However, it can be overcome by adjusting the differential pressure between cathode and anode carefully at the operational level. High gas permeability also leads to oxygen diffusion onto the cathode chamber and reduces the efficiency of the electrolyzer. This can be minimized by developing a membrane with low gas permeability which can limit the diffusion of oxygen to a certain level. In this regard, if we can reduce the pore diameter of the separator membrane then we can reduce the gas crossover which leads to reduced oxygen diffusion into the cathodic chamber and increases the efficiency of the electrolyzer [35, 36].

Point 2: The diffusion of oxygen into the cathode chamber reduces the efficiency of the electrolyser – how to overcome this issue?

Response 2: We are grateful for the comment. This is one of the very key issues which has been highlighted by the reviewer. In all types of electrolyser, whether it is Alkaline, PEM, or Anion exchange membrane electrolyser, diffusivity problem occurs at low current density [1-2]. The current efficiency decreased only when the current density is low at about 0.1~0.2 A cm-2 for all the electrolyzers such as PEM and ALK. [3]

[1] Journal of The Electrochemical Society, 165 (7) F502-F513 (2018).

[2I] International journal of hydrogen energy 38 14921e14933 (2013). 

[3] Journal of The Electrochemical Society, 163 (11) F3197-F3208 (2016).

To minimize this problem in the ALK system, it is important to improve the permeability property of the separator membrane. By developing a membrane with low gas permeability or high gas resistance we can limit the diffusion of oxygen to a certain level. In this regard, if we can reduce the pore diameter of the separator membrane then we can reduce the gas crossover which leads to reduced oxygen diffusion into the cathodic chamber and increases the efficiency of the electrolyser.

[Page 2] When the membrane suffered from high gas permeability then it causes high gas crossover issues and made the operation difficult at the high-pressure level. However, it can be overcome by adjusting the differential pressure between cathode and anode carefully at the operational level. High gas permeability also leads to oxygen diffusion onto the cathode chamber and reduces the efficiency of the electrolyzer. This can be minimized by developing a membrane with low gas permeability which can limit the diffusion of oxygen to a certain level. In this regard, if we can reduce the pore diameter of the separator membrane then we can reduce the gas crossover which leads to reduced oxygen diffusion into the cathodic chamber and increases the efficiency of the electrolyzer [35, 36].

Point 3:  The low gas crossover rate of the solid polymer electrolyte membrane allows for the PEM electrolyser to work under a wide range of power input; how this can be justified for AWE using liquid KOH electrolyte?

Response 3:  We thank the reviewer’s comment. According to “Journal of The Electrochemical Society, 165 (7) F502-F513 (2018)” PEM and AWE show a similar anodic hydrogen content as a function of current density in Fig. 3. The anodic hydrogen content of the AWE electrolyzer which is operated at 20 bars was even lower than that of the PEM electrolyzer at 20 bars when the separated electrolytes process was applied. This implied that the gas crossover can be controlled by process management strategies. The important thing is “The wide range of power output in case of PEM is attributed to higher current density values, not to the use of solid polymer electrolyte membrane.”

[Page 2] PEM exhibits high current density (2 A cm-2 @ ~2 V), low ohmic losses, large partial load range, flexibility, swift response, low overpotential and produce highly pure hydrogen. This high current density allows the PEM electrolyzer to work under a wide range of power output [16, 17].

Another reason for the low operating pressure and low partial load range is the non-negligible permeability of dissolved gas in the liquid electrolyte i.e., KOH through the separator membrane [33, 34]. When the membrane suffered from high gas permeability then it causes high gas crossover issues and made the operation difficult at the high-pressure level. However, it can be overcome by adjusting the differential pressure between cathode and anode carefully at the operational level. High gas permeability also leads to oxygen diffusion onto the cathode chamber and reduces the efficiency of the electrolyzer. This can be minimized by developing a membrane with low gas permeability which can limit the diffusion of oxygen to a certain level. In this regard, if we can reduce the pore diameter of the separator membrane then we can reduce the gas crossover which leads to reduced oxygen diffusion into the cathodic chamber and increases the efficiency of the electrolyzer [35, 36].

[Page 10] On the other hand, the polymer-based electrolyte membrane in the case of PEM electrolyzer normally consists of nanopores of a few nm around (2–5 nm), due to which they showed low crossover of gas induced by the differential pressure. To reduce the gas crossover induced by the difference of pressure, porous separator membranes with small size pores can be developed.

Point 4: How about using Nafion membrane? And having ternary mixtures of electrocatalysts?

Response 4:  We thank the reviewer’s comment. In the past, some effort was made to use Nafion membrane as a separator material for Alkaline water electrolyser. However, Nafion membrane exhibited poor performance in KOH electrolyte and gave very low water content, high cell resistance, and high cell potential which reduced the efficiency of Alkaline water electrolyser. Though in NaOH electrolyte, the Nafion membrane showed better performance as compared to KOH, the properties are not good in comparison to Commercial Zirfon and other membranes used for AWE [1-2].

Many studies published novel ternary mixtures of electrocatalysts, which will help to enhance the efficiency [3-4].

[1] Journal of Applied Electrochemistry volume 10, pages 741–747 (1980)

[2] Journal of Membrane Science Volume 493, 1 November 2015, Pages 589-598

[3] International Journal of Hydrogen Energy Volume 38, Issue 36, 13 December 2013, Pages 15928-15933

[4] Advanced Materials Volume 29, Issue 9, March 7, 2017, 1605502

[Page 3] In the past, some effort was made to use Nafion membrane as a separator material for Alkaline water electrolyser. However, Nafion membrane exhibited poor performance in KOH electrolyte and gave very low water content, high cell resistance, and high cell potential which reduced the efficiency of Alkaline water electrolyser. Though in NaOH electrolyte, the Nafion membrane showed better performance as compared to KOH, the properties are not good in comparison to Commercial Zirfon® and other membranes used for AWE [63, 64].

Point 5:  Please provide the composition for Raney nickel; ionic conductivity of the KOH electrolyte.

.

Response 5: We thank the reviewer’s comment. Raney-type materials are the alloys of electrocatalytically active metals like (Ni, Co, Cu) with readily leached metals like zinc and aluminum. Active metals can leach out easily in alkaline solutions. These materials are mostly used to enhance the real surface area, however, the surfaces belonging to the rough structures can also exhibit high electrocatalytic activity. Raney nickel used in the present study only contains nickel. The composition is added to the manuscript.

          The ionic conductivity of the KOH electrolyte used in this study was 0.4468 S/cm. The value is added in the text.

[Page 4]   Raney nickel and Nickel-Iron (Ni-Fe) LDH was employed on the cathode and anode side respectively as a catalyst. Raney-type materials are the alloys of electrocatalytically active metals like (Ni, Co, Cu) with readily leached metals like zinc and aluminum. Active metals can leach out easily in alkaline solutions. These materials are mostly used to enhance the real surface area, however, the surfaces belonging to the rough structures can also exhibit high electrocatalytic activity. Raney nickel used in the present study only contains nickel.

[Page 6]    cathode side was Raney nickel (composed of nickel only)

[Page 10]    The ionic conductivity of the KOH electrolyte used in this study was 0.4468 S/cm.

Point 6: In the last part of the introduction lines 80-82; A range of other inorganic fillers and their influence on ionic conductivity has also been reported in the literature, doi.org/10.1016/j.electacta.2014.06.039; doi.org/10.1021/ie502615w – please include and expand the discussion.

Response 6:  We thank the reviewer’s comment. We included the valuable literature work as advised by the respected reviewer. We revised the whole introduction and added new citations as advised. Some of them are mentioned here.

  1. Takach, M., M. Sarajlić, D. Peters, M. Kroener, F. Schuldt and K. von Maydell, Review of Hydrogen Production Techniques from Water Using Renewable Energy Sources and Its Storage in Salt Caverns. Energies, 2022. 15(4): p. 1415.
  2. Minakshi, M., S. Higley, C. Baur, D.R. Mitchell, R.T. Jones and M. Fichtner, Calcined chicken eggshell electrode for battery and supercapacitor applications. RSC Adv. 2019. 9(46): p. 26981-26995.
  3. Divakaran, A.M., D. Hamilton, K.N. Manjunatha and M. Minakshi, Design, development and thermal analysis of reusable Li-ion battery module for future mobile and stationary applications. Energies, 2020. 13(6): p. 1477.
  4. Wickramaarachchi, W.K.P., M. Minakshi, X. Gao, R. Dabare and K.W. Wong, Hierarchical porous carbon from mango seed husk for electro-chemical energy storage. Adv. Chem. Eng., 2021. 8: p. 100158.
  5. Minakshi, M., D.R. Mitchell, C. Baur, J. Chable, A.J. Barlow, M. Fichtner, A. Banerjee, S. Chakraborty and R. Ahuja, Phase evolution in calcium molybdate nanoparticles as a function of synthesis temperature and its electrochemical effect on energy storage. Nanoscale Adv. 2019. 1(2): p. 565-580.
  6. Minakshi, M., D.R. Mitchell, R.T. Jones, N.C. Pramanik, A. Jean‐Fulcrand and G. Garnweitner, A hybrid electrochemical energy storage device using sustainable electrode materials. ChemistrySelect, 2020. 5(4): p. 1597-1606.
  7. Sharma, P., M. Minakshi, J. Whale, A. Jean-Fulcrand and G. Garnweitner, Effect of the anionic counterpart: Molybdate vs. tungstate in energy storage for pseudo-capacitor applications. Nanomaterials, 2021. 11(3): p. 580.
  8. Divakaran, A.M., M. Minakshi, P.A. Bahri, S. Paul, P. Kumari, A.M. Divakaran and K.N. Manjunatha, Rational design on materials for developing next generation lithium-ion secondary battery. Prog. Solid. State Ch. 2021. 62: p. 100298.
  9. Verma, M.L., M. Minakshi and N.K. Singh, Synthesis and characterization of solid polymer electrolyte based on activated carbon for solid state capacitor. Electrochim. Acta, 2014. 137: p. 497-503.
  10. Verma, M.L., M. Minakshi and N.K. Singh, Structural and electrochemical properties of nanocomposite polymer electrolyte for electrochemical devices. Ind. Eng. Chem. Res. 2014. 53(39): p. 14993-15001.

Introduction:

The power to gas (PtG) technology can address the continuing and extensive energy storage issues along with minimization in CO2 emissions. PtG process changes electric power into chemical energy for steady storage of energy at great length. This system can be very useful in the betterment of energy-related systems in the future [1-4]. For the implementation of renewable energy on large scale, a cost-effective and energy-efficient water electrolysis system is required [5]. Like water electrolysis, energy storage devices also reduced the emission of greenhouse gases and mitigate the issues of global warming. Energy storage devices like batteries and capacitors are also a hot topic in the energy sector these days owing to their ease of energy storage and usage according to the preference of users. However, for energy storage devices the electrode design is very important and it's a very demanding job to design it [6-12].

Currently, hydrogen generation through water electrolysis is a suitable choice for the multi gigawatt storage of electrical energy from different irregular sources of energy like solar and wind. Water electrolysis can easily change surplus electricity into useful hydrogen with greater flexibility [13]. The future of renewable hydrogen production mainly depends on polymer exchange membrane (PEM) and alkaline water electrolysis (AWE) technologies [14, 15]. PEM exhibits high current density (2 A cm-2 @ ~2 V), low ohmic losses, large partial load range, flexibility, swift response, low overpotential and produce highly pure hydrogen. This high current density allows the PEM electrolyzer to work under a wide range of power output [16, 17]. However, it has some gas crossover issues which need to be addressed for better performance of electrolyzer [18]. Additionally, the use of expensive noble metal-type catalysts and titanium-based current collectors make this system less cost-effective [19, 20].  

On the contrary, alkaline water electrolyzers (AWEs) exhibit higher durability, simplicity, robust performance, low capital cost, and adequate compatibility with non-noble metal catalysts [21-25]. AWE is in commercial use for quite a long time and is considered a mature technology [26]. However, they exhibit low operating pressure and limited partial load range, slow response time towards dynamic operations, and most importantly low current densities (below 800 mA/cm2 @ 1.8 V), owing to the higher values of ionic resistance of the separator membrane and the lower kinetics caused by the utilization of non-noble type catalysts [27-29]. These lower densities increased the number of stacks required to obtain enough hydrogen [30-32]. Another reason for the low operating pressure and low partial load range is the non-negligible permeability of dissolved gas in the liquid electrolyte i.e., KOH through the separator membrane [33, 34]. When the membrane suffered from high gas permeability then it causes high gas crossover issues and made the operation difficult at the high-pressure level. However, it can be overcome by adjusting the differential pressure between cathode and anode carefully at the operational level. High gas permeability also leads to oxygen diffusion onto the cathode chamber and reduces the efficiency of the electrolyzer. This can be minimized by developing a membrane with low gas permeability which can limit the diffusion of oxygen to a certain level. In this regard, if we can reduce the pore diameter of the separator membrane then we can reduce the gas crossover which leads to reduced oxygen diffusion into the cathodic chamber and increases the efficiency of the electrolyzer [35, 36].

To overcome the issues related to gas crossover, significant research on alkaline anion exchange membranes (AEMs) were carried out, but they exhibited lower stability as compared to the acidic type membranes [37, 38]. Thus, a lot of notable work was done to decrease the ohmic losses by manufacturing much improved and advanced separator membranes and AEMs.

Commercial AWEs use porous diaphragm materials as separators because of their high durability. Currently, Agfa’s Zirfon® is used commercially as the diaphragm, Zirfon® is a combination of polysulfone (PSU) and ZrO2 nanoparticles. Zirconia being hydrophilic provides excellent wettability and stiffness to the separator while polysulfone is a binder and imparts flexibility. Zirfon® exhibits excellent stability in an aqueous KOH solution. Previously, asbestos was used as a separator membrane for AWE, however, due to its toxic nature, it was replaced. Additionally, Zirfon® exhibits superior properties as compared to asbestos [39-43]. Porous Zirfon® having its big pore size (up to ~130 nm), promotes the transport of gas-containing electrolyte across the separator resulting in an increased electrical conductivity as well as gas crossover [44]. This increased gas crossover resulted in a significant reduction in the dynamic range of the electrolyzer [45, 46]. Gas crossover can be reduced by increasing the thickness of the porous separator membrane; however, it will also increase the ohmic voltage drop in the electrolyte resulting in lower energy efficiency [47]. For that reason, the need of the hour is to develop advanced separators membranes having excellent ionic conductivity besides the reduced gas crossover for highly efficient AWEs. 

Several studies were being conducted to manufacture a film with improved ionic conductivity while controlling the pore size to be small. In recent times, a lot of work has been done on thermoplastic nature-based polymer separators owing to the excellent stability and better handling properties of this type of polymer. However, owing to the hydrophobic behavior of these polymers, inorganic hydrophilic filler materials are added to enhance the surface properties of the separator. Polysulfone (PSU) is normally applied as a membrane material owing to its high thermal, chemical, and mechanical stabilities [48, 49]. Therefore, the use of PSU as matrix material is very favorable to get higher chemical stability under harsh alkaline environments [50]. Many techniques have been developed to improve the surface characteristics of polysulfone like sulfonation [51], crosslinking [52], or blending [53]. Several inorganic materials like ZrO2 [54, 55], CeO2 [56], TiO2 [57, 58], yttria-stabilized zirconia commonly known as (YSZ) [59], barite mineral (BaSO4) have been added as a filler in the polymer matrix [60]. However, these filler materials exhibited a significant reduction in their mechanical stabilities due to agglomeration and improper mixing with polymer matrix [45, 61, 62]. In the past, some effort was made to use Nafion membrane as a separator material for Alkaline water electrolyser. However, Nafion membrane exhibited poor performance in KOH electrolyte and gave very low water content, high cell resistance, and high cell potential which reduced the efficiency of Alkaline water electrolyser. Though in NaOH electrolyte, the Nafion membrane showed better performance as compared to KOH, the properties are not good in comparison to Commercial Zirfon® and other membranes used for AWE [63, 64].

There exists a great influence of inorganic fillers on ionic conductivities. For this reason, a novel approach was adopted in the literature in which silver ion conducting solid polymer electrolyte was incorporated with activated carbon which demonstrates high ionic conductivities and acted as a potential choice for energy storage devices for instance solid-state capacitor applications [65]. Nanocomposite’s polymer electrolytes (silica as inorganic filler material incorporated in the polymer matrix) showed a certain effect on the ionic conductivities. The concentration of filler in a polymer matrix is very important for the ionic conductivities which affect the performance of energy storage devices [66].

Several membranes were used previously for alkaline water electrolysis systems. Polyvinyl alcohol-based separator membranes exhibited good thermal stability and adequate mechanical strength [57]. Separator membranes containing different mineral fillers like BaSO4 were also employed and gave slightly good resistivity in comparison to asbestos-based membranes [60, 67]. Alkali-doped polyvinyl alcohol polybenzimidazole-based membranes were also utilized for AWE. These membranes indicate good thermal and chemical stabilities in aqueous KOH solution however, they were expensive [68]. Additionally, these separator membranes exhibited reduced performance in comparison to Zirfon®.

In the present work, we managed to synthesize highly ionic conductive zirconia toughened alumina (ZTA) (with different Al2O3 to ZrO2 ratio) based porous composite separator membrane for alkaline electrolyzer via phase inversion process. The compatibility of filler material (ZTA) with the polymer was tested. The ionic resistance and gas permeability of ZTA based composite separator membranes were observed by using KOH electrolyte. The electrolysis of prepared separators was conducted by changing the temperature during electrolysis and varying the amount and flow rate of KOH electrolyte solution.

Point 7: In the results and discussion – Text relating Figure 4b (lines 316 -319); Nyquist plot of impedance spectra has not been explained well. Please provide ohmic resistance values and explain the shape of the diameter of the semicircles. Some electrochemical insights on the Nyquist plot (using the background cited in the literature 10.1039/C8NR03824D) will assist the readers to understand the impedance and its role on the mass transport aspects and ohmic resistances.

Response 7:  We thank the reviewer’s comment. We revised the text in the manuscript and provided the related information asked by the respected reviewer.

[Page 13] Figure 8b shows the Nyquist plot of impedance spectra at 1 A/cm2. The equivalent circle largely consists of ohmic resistance, activation resistance, mass-transport resistance. The high-frequency resistance (HFR) seems like the intercepts of the Nyquist plot with the x-axis at higher frequency (left portion on Nyquist plot), representing the ohmic resistance of the cell. The ohmic resistance is the resistance caused by the flow of current through the cell, which is mainly the contributions from the membrane. The area resistance measure ex-situ in Figure 3b matched well with the in-situ internal resistance of Figure 8b. The middle portion of the equivalent circuit model exhibits the activation loss. The activation loss results from the kinetics of electrodes. The only one semicircle from high to the middle frequency with a similar size has been observed for all the membrane separators, which is attributed to the use of the same electrodes. The mass-transport limitation and resistance were not observed in this study.

Point 8: In Fig. 5c how the cell voltage has been as high as 3.6 V while using aqueous solutions? Justify.

Response 8: The reviewer raised an important question. The Zirfon® exhibits high cell voltage 3.6 V at higher current density 2 A/cm2 values owing to the use of diluted aqueous KOH solution of 10 wt.%. It is well known that the ionic conductivity values decreased with the lower KOH concentration. This result implies that the internal resistance of the Zirfon separator is relatively higher than our prepared membranes.

Journal of The Electrochemical Society, 163 (11) F3125-F3131 (2016).

[Page 14] KOH concentration from 10 wt. % to 30 wt. %. The Zirfon® exhibits high cell voltage 3.6 V at high current density 2 A/cm2 values owing to the use of diluted aqueous KOH solution of 10 wt.%. It is well known that ionic conductivity values decreased with lower KOH concentration, which implies that the internal resistance of Zirfon® is relatively higher than our prepared membranes.

Reviewer 2 Report

The article by Won-Chul Cho and colleagues addresses a very important problem of developing a porous separator membrane for the process of alkaline water electrolysis. The authors, based on a commercially available Zirfon separator membrane, have prepared two samples of new zirconia toughened alumina based separator membranes. These samples were examined by various methods, including a stability test and better properties of one of them compared to Zirfon membrane were shown. I think that the article is well developed; the results are presented quite clearly and the article can be published in the Polymers after some revision.

  1. First five sentences in the abstract seem to be more proper for introduction part.
  2. The topic of searching for a membrane for hydrogen production by electrolysis is one of the most studied at the moment, so it is surprising that the authors in the introduction part cited the references to relatively old works, whereas many works on this topic have been published recently, including review articles. Could the authors update the references in the introduction part, if possible?
  3. The part 2.2 is written too briefly. Could you write a synthesis methodology in more detail, specifying the amount of substances to be loaded, or give a reference to a similar methodology?
  4. I couldn't find the supplementary materials if there were to be any. The file was not attached; the link at the end of the article does not work.
  5. General remark. There are quite a few typos in the text, including repetitions of words (line 130), unsuccessful expressions (lines 102, 151, 200-201, 299), unedited formulas (lines 32, 81, 86, 217, 218, 220, 222, 366, 367) and name of substances (line 97 – should be N-methyl-2-pyrrolidone or N-methyl-2-pyrrolidinone). There are also problems with spaces (for example lines 313, 325 and so on).

Author Response

Response to Reviewer 2 Comments

Points: The article by Won-Chul Cho and colleagues addresses a very important problem of developing a porous separator membrane for the process of alkaline water electrolysis. The authors, based on a commercially available Zirfon separator membrane, have prepared two samples of new zirconia toughened alumina-based separator membranes. These samples were examined by various methods, including a stability test and better properties of one of them compared to Zirfon membrane were shown. I think that the article is well developed; the results are presented quite clearly, and the article can be published in Polymers after some revision.

Response:  We are grateful to the reviewer for his/her careful review of our manuscript and constructive comments.

Point 1: First five sentences in the abstract seem to be more proper for the introduction part.

Response 1:  We are grateful to the reviewer for the comment. The abstract is modified in the text as advised and reproduced here.

[Page 1] Hydrogen is nowadays considered as a favorable and attractive energy carrier fuel to replace the sectors causing global warming problems. Water electrolysis has attained great attention from researchers to produce green hydrogen mainly for the accumulation of renewable energy. Hydrogen can be safely used as a bridge to successfully connect the energy demand and supply division. An alkaline water electrolysis system owing to its low cost can efficiently use renewable energy sources on large scale. Normally organic/inorganic composite porous separator membranes have been employed as a membrane for alkaline water electrolyzers. However, the separator membranes exhibit high ionic resistance and low gas resistance values, resulting in lower efficiency and raised safety issues as well. Here, in this study, we report that zirconia toughened alumina (ZTA)-based separator membrane exhibits less ohmic resistance 0.15 ٞcm2 and low hydrogen gas permeability 10.7 × 10-12 mol cm-1 sec-1 bar-1 in 30 wt.% KOH solution, which outperforms the commercial state of the art Zirfon® PERL separator. The cell containing ZTA, and advanced catalysts exhibit the excellent performance of 2.1 V at 2000 mA/cm2 at 30 wt.% KOH and 80 °C, which is comparable with PEM electrolysis. These improved results show that AWEs equipped with ZTA separators could be superior to PEM electrolysis performance.

Point 2: The topic of searching for a membrane for hydrogen production by electrolysis is one of the most studied at the moment, so it is surprising that the authors in the introduction part cited the references to relatively old works, whereas many works on this topic have been published recently, including review articles. Could the authors update the references in the introduction part, if possible?

Response 2:  We are grateful to the reviewer for the comment. The Introduction part is modified, and references are updated in the manuscript as advised. A few of the latest referenced articles are mentioned here:

  1. Takach, M., M. Sarajlić, D. Peters, M. Kroener, F. Schuldt and K. von Maydell, Review of Hydrogen Production Techniques from Water Using Renewable Energy Sources and Its Storage in Salt Caverns. Energies, 2022. 15(4): p. 1415.
  2. Sharma, P., M. Minakshi, J. Whale, A. Jean-Fulcrand and G. Garnweitner, Effect of the anionic counterpart: Molybdate vs. tungstate in energy storage for pseudo-capacitor applications. Nanomaterials, 2021. 11(3): p. 580.
  3. Divakaran, A.M., M. Minakshi, P.A. Bahri, S. Paul, P. Kumari, A.M. Divakaran and K.N. Manjunatha, Rational design on materials for developing next generation lithium-ion secondary battery. Prog. Solid. State Ch. 2021. 62: p. 100298.
  4. David, M., C. Ocampo-Martínez and R. Sánchez-Peña, Advances in alkaline water electrolyzers: A review. J. Energy Storage, 2019. 23: p. 392-403.
  5. Reier, T., H.N. Nong, D. Teschner, R. Schlögl and P. Strasser, Electrocatalytic oxygen evolution reaction in acidic environments–reaction mechanisms and catalysts. Adv. Energy Mater. 2017. 7(1): p. 1601275.
  6. Stelmachowski, P., J. Duch, D. Sebastián, M.J. Lázaro and A. Kotarba, Carbon-based composites as electrocatalysts for oxygen evolution reaction in alkaline media. Materials, 2021. 14(17): p. 4984.
  7. Abbasi, R., B.P. Setzler, S. Lin, J. Wang, Y. Zhao, H. Xu, B. Pivovar, B. Tian, X. Chen and G. Wu, A Roadmap to Low‐Cost Hydrogen with Hydroxide Exchange Membrane Electrolyzers. Adv. Mater. 2019. 31(31): p. 1805876.
  8. Guilbert, D. and G. Vitale, Hydrogen as a Clean and Sustainable Energy Vector for Global Transition from Fossil-Based to Zero-Carbon. Clean Technol. 2021. 3(4): p. 881-909.
  9. Yodwong, B., D. Guilbert, M. Phattanasak, W. Kaewmanee, M. Hinaje and G. Vitale, AC-DC converters for electrolyzer applications: State of the art and future challenges. Electronics, 2020. 9(6): p. 912.
  10. Kim, S., J.H. Han, J. Yuk, S. Kim, Y. Song, S. So, K.T. Lee and T.-H. Kim, Highly selective porous separator with thin skin layer for alkaline water electrolysis. J. Power Sources, 2022. 524: p. 231059.
  11. Brauns, J., J. Schönebeck, M.R. Kraglund, D. Aili, J. Hnát, J. Žitka, W. Mues, J.O. Jensen, K. Bouzek and T. Turek, Evaluation of diaphragms and membranes as separators for alkaline water electrolysis. J. Electrochem. Soc. 2021. 168(1): p. 014510.

Introduction:

The power to gas (PtG) technology can address the continuing and extensive energy storage issues along with minimization in CO2 emissions. PtG process changes electric power into chemical energy for steady storage of energy at great length. This system can be very useful in the betterment of energy-related systems in the future [1-4]. For the implementation of renewable energy on large scale, a cost-effective and energy-efficient water electrolysis system is required [5]. Like water electrolysis, energy storage devices also reduced the emission of greenhouse gases and mitigate the issues of global warming. Energy storage devices like batteries and capacitors are also a hot topic in the energy sector these days owing to their ease of energy storage and usage according to the preference of users. However, for energy storage devices the electrode design is very important and it's a very demanding job to design it [6-12].

Currently, hydrogen generation through water electrolysis is a suitable choice for the multi gigawatt storage of electrical energy from different irregular sources of energy like solar and wind. Water electrolysis can easily change surplus electricity into useful hydrogen with greater flexibility [13]. The future of renewable hydrogen production mainly depends on polymer exchange membrane (PEM) and alkaline water electrolysis (AWE) technologies [14, 15]. PEM exhibits high current density (2 A cm-2 @ ~2 V), low ohmic losses, large partial load range, flexibility, swift response, low overpotential and produce highly pure hydrogen. This high current density allows the PEM electrolyzer to work under a wide range of power output [16, 17]. However, it has some gas crossover issues which need to be addressed for better performance of electrolyzer [18]. Additionally, the use of expensive noble metal-type catalysts and titanium-based current collectors make this system less cost-effective [19, 20].  

On the contrary, alkaline water electrolyzers (AWEs) exhibit higher durability, simplicity, robust performance, low capital cost, and adequate compatibility with non-noble metal catalysts [21-25]. AWE is in commercial use for quite a long time and is considered a mature technology [26]. However, they exhibit low operating pressure and limited partial load range, slow response time towards dynamic operations, and most importantly low current densities (below 800 mA/cm2 @ 1.8 V), owing to the higher values of ionic resistance of the separator membrane and the lower kinetics caused by the utilization of non-noble type catalysts [27-29]. These lower densities increased the number of stacks required to obtain enough hydrogen [30-32]. Another reason for the low operating pressure and low partial load range is the non-negligible permeability of dissolved gas in the liquid electrolyte i.e., KOH through the separator membrane [33, 34]. When the membrane suffered from high gas permeability then it causes high gas crossover issues and made the operation difficult at the high-pressure level. However, it can be overcome by adjusting the differential pressure between cathode and anode carefully at the operational level. High gas permeability also leads to oxygen diffusion onto the cathode chamber and reduces the efficiency of the electrolyzer. This can be minimized by developing a membrane with low gas permeability which can limit the diffusion of oxygen to a certain level. In this regard, if we can reduce the pore diameter of the separator membrane then we can reduce the gas crossover which leads to reduced oxygen diffusion into the cathodic chamber and increases the efficiency of the electrolyzer [35, 36].

To overcome the issues related to gas crossover, significant research on alkaline anion exchange membranes (AEMs) were carried out, but they exhibited lower stability as compared to the acidic type membranes [37, 38]. Thus, a lot of notable work was done to decrease the ohmic losses by manufacturing much improved and advanced separator membranes and AEMs.

Commercial AWEs use porous diaphragm materials as separators because of their high durability. Currently, Agfa’s Zirfon® is used commercially as the diaphragm, Zirfon® is a combination of polysulfone (PSU) and ZrO2 nanoparticles. Zirconia being hydrophilic provides excellent wettability and stiffness to the separator while polysulfone is a binder and imparts flexibility. Zirfon® exhibits excellent stability in an aqueous KOH solution. Previously, asbestos was used as a separator membrane for AWE, however, due to its toxic nature, it was replaced. Additionally, Zirfon® exhibits superior properties as compared to asbestos [39-43]. Porous Zirfon® having its big pore size (up to ~130 nm), promotes the transport of gas-containing electrolyte across the separator resulting in an increased electrical conductivity as well as gas crossover [44]. This increased gas crossover resulted in a significant reduction in the dynamic range of the electrolyzer [45, 46]. Gas crossover can be reduced by increasing the thickness of the porous separator membrane; however, it will also increase the ohmic voltage drop in the electrolyte resulting in lower energy efficiency [47]. For that reason, the need of the hour is to develop advanced separators membranes having excellent ionic conductivity besides the reduced gas crossover for highly efficient AWEs. 

Several studies were being conducted to manufacture a film with improved ionic conductivity while controlling the pore size to be small. In recent times, a lot of work has been done on thermoplastic nature-based polymer separators owing to the excellent stability and better handling properties of this type of polymer. However, owing to the hydrophobic behavior of these polymers, inorganic hydrophilic filler materials are added to enhance the surface properties of the separator. Polysulfone (PSU) is normally applied as a membrane material owing to its high thermal, chemical, and mechanical stabilities [48, 49]. Therefore, the use of PSU as matrix material is very favorable to get higher chemical stability under harsh alkaline environments [50]. Many techniques have been developed to improve the surface characteristics of polysulfone like sulfonation [51], crosslinking [52], or blending [53]. Several inorganic materials like ZrO2 [54, 55], CeO2 [56], TiO2 [57, 58], yttria-stabilized zirconia commonly known as (YSZ) [59], barite mineral (BaSO4) have been added as a filler in the polymer matrix [60]. However, these filler materials exhibited a significant reduction in their mechanical stabilities due to agglomeration and improper mixing with polymer matrix [45, 61, 62]. In the past, some effort was made to use Nafion membrane as a separator material for Alkaline water electrolyser. However, Nafion membrane exhibited poor performance in KOH electrolyte and gave very low water content, high cell resistance, and high cell potential which reduced the efficiency of Alkaline water electrolyser. Though in NaOH electrolyte, the Nafion membrane showed better performance as compared to KOH, the properties are not good in comparison to Commercial Zirfon® and other membranes used for AWE [63, 64].

There exists a great influence of inorganic fillers on ionic conductivities. For this reason, a novel approach was adopted in the literature in which silver ion conducting solid polymer electrolyte was incorporated with activated carbon which demonstrates high ionic conductivities and acted as a potential choice for energy storage devices for instance solid-state capacitor applications [65]. Nanocomposite’s polymer electrolytes (silica as inorganic filler material incorporated in the polymer matrix) showed a certain effect on the ionic conductivities. The concentration of filler in a polymer matrix is very important for the ionic conductivities which affect the performance of energy storage devices [66].

Several membranes were used previously for alkaline water electrolysis systems. Polyvinyl alcohol-based separator membranes exhibited good thermal stability and adequate mechanical strength [57]. Separator membranes containing different mineral fillers like BaSO4 were also employed and gave slightly good resistivity in comparison to asbestos-based membranes [60, 67]. Alkali-doped polyvinyl alcohol polybenzimidazole-based membranes were also utilized for AWE. These membranes indicate good thermal and chemical stabilities in aqueous KOH solution however, they were expensive [68]. Additionally, these separator membranes exhibited reduced performance in comparison to Zirfon®.

In the present work, we managed to synthesize highly ionic conductive zirconia toughened alumina (ZTA) (with different Al2O3 to ZrO2 ratio) based porous composite separator membrane for alkaline electrolyzer via phase inversion process. The compatibility of filler material (ZTA) with the polymer was tested. The ionic resistance and gas permeability of ZTA based composite separator membranes were observed by using KOH electrolyte. The electrolysis of prepared separators was conducted by changing the temperature during electrolysis and varying the amount and flow rate of KOH electrolyte solution.

Point 3: The part 2.2 is written too briefly. Could you write a synthesis methodology in more detail, specifying the amount of substances to be loaded, or give a reference to a similar methodology?

Response 3:  We are grateful to the reviewer for the comment. Part 2.2 is improved as advised and added in the manuscript. 

[Page 4] ZTA/PSU-based composite separator (ZTA 85 wt.% and Polysulfone 15 wt.%) was synthesized by using film casting technique. Firstly, NMP (which was used as a solvent) 112.19 g and PVP (additive) 4.479 g were placed in a mixing device (RED 150-D, Pendraulik) and mixed for a 40-minute time at 3000 revolutions per minute (RPM) value. After that PSU 16.76 g and ZTA nanoparticles, 95 g were also added to the mixture, and the slurry was mixed continuously at 3000 RPM to get a homogeneous suspension having required viscosity. The mixing was done at 40 °C. After preparing a homogeneous and stable suspension this suspension was then placed on a quartz plate and casting was done by applying a doctor blade at 40 °C. PPS mesh was added during the casting as supporting material. By using a doctor blade of different heights w.r.t the quartz plate, the separator’s thickness was altered. The prepared sample was then placed in an oven for drying at 80 °C for 15 minutes after that it was introduced in the coagulation bath for extraction. The extracted membrane sample was then stored in deionized water.

Point 4: I couldn't find the supplementary materials if there were to be any. The file was not attached; the link at the end of the article does not work.

Response 4:  We are grateful to the reviewer for the comment. There is no supplementary material for this article.  

Point 5: General remark. There are quite a few typos in the text, including repetitions of words (line 130), unsuccessful expressions (lines 102, 151, 200-201, 299), unedited formulas (lines 32, 81, 86, 217, 218, 220, 222, 366, 367) and name of substances (line 97 – should be N-methyl-2-pyrrolidone or N-methyl-2-pyrrolidinone). There are also problems with spaces (for example lines 313, 325 and so on).

Response 5: We are grateful for highlighting the mistakes and are rectified in the manuscript. Some of them are mentioned here

[Page 5] at room temperature

[Page 3] N-methyl-2-pyrrolidone (NMP, 99.9%)

[Page 8] Figure 3. (a) Mercury incremental intrusion volume curves of Zirfon®

[Page 1] minimization in CO2 emissions

[Page 3] barite mineral (BaSO4)

        different Al2O3 to ZrO2 ratio

[Page 8] top of the Z5TA95

            and Z30TA70 separator membranes

            The Z5TA95 which has very low

             like in Z30TA70

[Page 13] This makes Z30TA70 more

[Page 13] improved Z30TA70 separator

[Page 15] The polarization characteristics of Z30TA70

              Z30TA70 also improved  
